# The Chicken MHC: Insights into Genetic Resistance, Immunity, and Inflammation Following Infectious Bronchitis Virus Infections

**DOI:** 10.3390/vaccines8040637

**Published:** 2020-11-02

**Authors:** Ana P. da Silva, Rodrigo A. Gallardo

**Affiliations:** Department of Population Health and Reproduction, School of Veterinary Medicine, University of California, Davis, CA 95616, USA; apdasilva@ucdavis.edu

**Keywords:** chicken MHC, genetic resistance, IBV, innate immunity, inflammation

## Abstract

The chicken immune system has provided an immense contribution to basic immunology knowledge by establishing major landmarks and discoveries that defined concepts widely used today. One of many special features on chickens is the presence of a compact and simple major histocompatibility complex (MHC). Despite its simplicity, the chicken MHC maintains the essential counterpart genes of the mammalian MHC, allowing for a strong association to be detected between the MHC and resistance or susceptibility to infectious diseases. This association has been widely studied for several poultry infectious diseases, including infectious bronchitis. In addition to the MHC and its linked genes, other non-MHC loci may play a role in the mechanisms underlying such resistance. It has been reported that innate immune responses, such as macrophage function and inflammation, might be some of the factors driving resistance or susceptibility, consequently influencing the disease outcome in an individual or a population. Information about innate immunity and genetic resistance can be helpful in developing effective preventative measures for diseases such as infectious bronchitis, to which a systemic antibody response is often not associated with disease protection. In this review, we summarize the importance of the chicken MHC in poultry disease resistance, particularly to infectious bronchitis virus (IBV) infections and the role played by innate immunity and inflammation on disease outcome. We highlight how future studies focusing on the MHC and non-MHC genes can potentially bring clarity to observed resistance in some chicken B haplotype lines.

## 1. Introduction

The avian immune system has been studied for decades and has provided immense knowledge on fundamental concepts in basic immunology that are widely accepted today. Despite numerous similarities to the mammalian immune system [1], avian species have unique immunological structures and mechanisms. For example, the serendipitous discovery of the bursa of Fabricius as the site for B cell antigen-specific repertoire development led to the differentiation between B and T cells, which are now identified as the two arms of adaptive immunity [2]. Unlike mammals, lymph nodes are absent in the chicken, although lymphoid tissue aggregates are distributed in several organs and systems. Some examples of these lymphoid tissues include lymphoid nodules in the walls of lymphatic vessels, the gut-, mucosa-, bronchial- and conjunctiva-associated lymphoid tissues (GALT, MALT, BALT, and CALT), cecal tonsils, Meckel’s diverticulum, Peyer’s patches, and Harderian glands [3]. These specific features of the chicken immune system and the importance of improving health in poultry production have turned the chicken into an important animal model for research, aiding the understanding of several aspects of basic immunology [4]. A special landmark of avian immunology was the unforeseen attenuation of a pathogen, later named *Pasteurella multocida*, by Louis Pasteur when studying fowl cholera in the late 1800s. His breakthrough led to the discovery of the first attenuated vaccine [5]. Some vaccine delivery systems that are inconvenient or inappropriate for mammalian livestock are widely used in poultry. These systems include mucosal vaccination via post-hatch spray and eye drop vaccines, which stimulate local innate and adaptive immune responses originating in the Harderian glands and local MALT, and in ovo vaccination, which enables easy automation and early stimulation of adaptive immune responses during embryonic development [3].

In addition to humoral responses elicited by natural infections and vaccination, cell-mediated adaptive immune responses in birds play an important role in minimizing disease outcome and preventing reinfections. For example, cytotoxic cluster of differentiation 8-positive (CD8^+^) T cells play a major role in eliminating infectious bronchitis virus (IBV) [6]. In addition, helper CD4^+^ T cells become activated by IBV antigens displayed on the surface of antigen presenting cells. After activation, CD4^+^ T cells interact with other T and B cells, amplifying cytotoxic and humoral responses to IBV [7]. Moreover, innate cellular responses also assist in fighting early stages of viral infections. The rates of macrophage differentiation and activation have been shown to be greater in relatively IBV-resistant cells than in IBV-susceptible cells, suggesting that disease resistance might be linked to a vigorous innate immune response [8,9,10]. Cytokine production and proinflammatory responses induced by IBV might also affect the severity of the disease and the development of appropriate adaptive immune responses [11,12,13].

Another remarkable feature of the chicken immune system is the major histocompatibility complex (MHC). Unlike the MHC of mammals, the chicken MHC is compact and simple, while still maintaining the essential counterpart genes. The chicken MHC contains about 46 genes in a region of about 209 kb of the chicken genome [14], whereas the human MHC contains over 200 genes spanning a region of approximately 4000 kb [15]. Moreover, there is a single dominantly expressed chicken MHC class I antigen, which allows detection of a strong association between the chicken MHC and resistance or susceptibility to infectious diseases (Figure 1) [16,17]. The genetic makeup of the host determines how the immune system will respond to infectious challenges and, ultimately, if the response will be protective or not. Within a population, the genetic variability of the MHC and other related genes leads to a wide range of immune responses and disease outcomes, varying from mild clinical signs to mortality. This variability is partly due to the polymorphic nature of genes that regulate the expression of components of the immune system [18].

To study disease resistance in chickens, MHC congenic chicken lines that share the same genetic background with differences exclusively in their MHC B locus have been developed [20,21,22,23]. Using these chicken lines as animal models, associations between MHC haplotypes and disease resistance or susceptibility have been described for several infectious pathogens, including coccidia [24,25], pathogenic bacteria [26,27,28,29], oncogenic viruses [30,31,32,33,34,35,36,37], and other viruses including IBV [38,39,40,41,42,43,44,45,46,47]. The main focus of the present review is IBV.

Disease prevention and control are of major concern in commercial poultry production. Vaccination, in addition to management and biosecurity measures, is widely used to prevent disease and control pathogen loads when eradication is not possible. Vaccines, however, rarely provide 100% protection against infectious diseases [48]. With some diseases, such as infectious bronchitis, high serum antibody levels elicited by vaccination do not necessarily correspond to protection [46,49,50]. Studying the genetic resistance to IBV determined by the chicken MHC is essential to better understand the mechanisms underlying resistance and, consequently, to use this knowledge to improve IBV prevention [6,9,12,39,42,44,46,47].

This review summarizes the significance of the chicken MHC and its impact on poultry disease resistance, particularly in IBV infections, and the future of IBV genetic resistance research using MHC inbred and congenic chicken lines.

## 2. The Chicken MHC 

The MHC is a well-studied genomic region of many animal species. It was first described in mice as the genetic locus responsible for rapid tissue allograft rejection and for encoding highly polymorphic alloantigens on cell surfaces [19,51]. In addition to its relation to graft rejection and autoimmunity, the MHC is also responsible for antigen presentation to T cells, representing an important bridge between the innate and the adaptive immune response. MHC class I molecules are present on nucleated cells and responsible for presenting peptides to CD8^+^ T cells. In contrast, the MHC class II molecules are only present on antigen presenting cells (dendritic cells, macrophages, and B cells) and are able to activate CD4^+^ T cells [52]. The mammalian MHC is polygenic, consisting of numerous genes, pseudogenes, and repetitive paralogous regions located on different chromosomes. The mammalian MHC is also polymorphic, with multiple alleles of each gene within a population [52,53]. Thus, the mammalian MHC is large and complex, with distantly related genes that provide extensive diversity to its antigen-presenting glycoproteins. In contrast, the chicken MHC is small and simple, yet contains the essential counterparts of genes present in the mammalian MHC. For this reason, the chicken MHC is considered a minimal essential set of genes, despite the differences from mammalian MHCs in organization and structure [16].

Upon discovery, the chicken MHC was classified as the B blood group or B locus, coding for agglutination factors present on the surface of chicken red blood cells [54]. Subsequently, these genes were associated with skin graft rejection and, therefore, histocompatibility [55]. Ultimately, it was discovered that the chicken MHC B locus is located on chicken microchromosome 16 and codes for molecules termed B-F (class I) and B-L (class II), which are closely linked and often referred to as the B-F/B-L region, and B-G (class IV), which is limited to Aves [56,57,58]. In addition to the B locus, a separate group of nonclassical MHC class I and II genes was identified and named Rfp-Y, which is also on microchromosome 16 but genetically distant and unlinked to the B locus (Figure 2) [59,60,61].

In addition to the MHC, genetic resistance to infectious diseases has been associated with CD1 genes, which are located near the B locus on microchromosome 16. CD1 glycoproteins are present on the surface of antigen-presenting cells and are responsible for detecting lipids and glycolipids and presenting them to specific subsets of T cells [62].

As previously mentioned, the chicken MHC is small and simple while still preserving essential homologous genes present in the mammalian MHC [16]. Furthermore, a single chicken MHC class I is predominantly expressed, showing a strong association with relative protection or vulnerability to various infectious diseases [19,62]. This genetic resistance to disease is demonstrated with specific MHC B haplotypes and, less frequently, with alleles in particular subregions of microchromosome 16 [62]. For this reason, the chicken is an excellent animal model to study immunology, leading geneticists to develop genetically defined chicken lines for research purposes. By using inbreeding, congenic chicken lines that share the same genetic backbone and differ from one another with respect to only the MHC B locus have been generated. With these chicken lines, scientists have been able to mitigate background effects, isolate and characterize specific genes, and associate MHC genes with disease resistance [20,21,22,23].

## 3. Disease Resistance in Chickens

The role of the MHC in disease resistance to a variety of infectious diseases caused by parasites, bacteria, and viruses has been studied extensively using MHC B congenic chicken lines. In this section, significant studies on noteworthy diseases are reviewed.

### 3.1. Parasites and Protozoa

Studies have shown an association between MHC haplotypes and increased levels of resistance to Northern fowl mite infestations. This resistance has been associated with skin inflammation that limits access to and feeding of mites on the host’s blood [63]. Using challenge experiments, it was observed that the intensity of the skin inflammation is influenced by the MHC genotype, and B21 haplotype chickens were described as more resistant to infestation than B15 haplotype chickens [64].

The severity of *Ascaridia* sp. infestations has also been associated with the MHC in chickens [65,66]. In addition, the MHC has been correlated to antibody titers against *Ascaridia* sp. However, although high antibody titers influenced the reduction in severity of the infestation, antibody titers were not correlated with ascariasis clearance [67].

Genetic effects of the MHC locus have been linked to susceptibility to coccidiosis [24,68,69]. An experiment in junglefowl populations demonstrated that a specific haplotype (CD-c), when homozygous, increased the host susceptibility to coccidiosis. However, susceptibility effects were masked by the second haplotype in heterozygotes, demonstrating the beneficial effects of heterozygosity in populations facing coccidian infections [70].

### 3.2. Bacterial Infections

In a field observational study comparing lame and clinically healthy broiler breeders, the presence of bacterial pathogens and the MHC haplotypes of the birds were analyzed and compared. Bacteria were isolated from 94.3% of lame birds, with *Staphylococcus* spp. being predominant. Homozygous B4 and B12 birds were more susceptible to bacterium-induced lameness than other homozygous haplotypes and to heterozygotes [71].

*Salmonella* sp. infections are a major public health concern in poultry production. Preventing salmonellosis and minimizing bacterial shedding is, therefore, of major interest to the industry. Different MHC haplotypes have been shown to contribute to differential resistance to *Salmonella* Enteritidis (SE), with haplotype BC appearing to be relatively resistant while B18 and B15 were relatively susceptible to SE-induced mortality [28]. Both MHC class I and II genes are associated with resistance to *Salmonella* sp. [29,72] and antibody kinetics [73].

Infections with *Escherichia coli* induce a wide range of pathologies in chickens. When inoculated subcutaneously, B13 haplotype chickens showed resistance to cellulitis compared with B21. Heterozygous B13/B21 chickens were more resistant to cellulitis than B21 birds [74]. On the other hand, intratracheal inoculation of *E. coli* in homozygous B21 haplotype layers exhibited more resistance to disease than B19/B21 birds. The differences in outcomes with B21 haplotype chickens might be due to the different genetic background of the birds used in these two experiments, given that resistance is a polygenic phenomenon and not exclusive to the B locus [75].

### 3.3. Oncogenic Viruses

Marek’s disease resistance studies date back to the late 1960s [76]. In early investigations, MHC B haplotypes B2, B19, B21, and BQ showed resistance to tumors induced by challenges with JM [35,77], GA-5, and RB-1B strains of Marek’s disease virus (MDV) [35]. After these initial discoveries, several studies have contributed to the understanding of the effects of MHC haplotypes in MDV infection and the different disease outcomes in chickens. In addition to inherent genetic disease resistance, the relationship between the MHC and immune response to MDV vaccines has been studied. Responses to MDV vary depending on vaccine serotypes and MHC haplotypes [78,79,80,81,82]. For example, B5 and B21 haplotypes have demonstrated varied protection levels when using MDV serotype 1, 2, and 3 vaccines, implicating that tailored vaccine strains should be used when certain MHC haplotypes are predominant in a population [79]. In addition, non-MHC loci have been studied by using recombinant congenic strains of chickens. These crosses are used to isolate specific subsets of genes, such as antibody allotypes and Rfp-Y genes, among others [83]. Eventually, the mapping of some genes located in chromosome 16 elucidated possible genes that could be linked to the development of Marek’s disease tumors, such as tripartite motif-containing protein genes (TRIM), butyrophilin gene (BTN), C-type lectin gene, and the Aves-specific B-G genes [84].

The first studies associating the chicken MHC B haplotype and resistance with Rous sarcoma virus (RSV) were performed in the 1970s [30,31]. While the B2 haplotype seemed to suppress the oncogenic effect of the viral v-src gene, the B5 haplotype chicken line seemed to be more permissive to tumor progression [85]. Later on, experiments were performed using recombinant congenic chicken lines to determine which genes were associated with Rous sarcoma tumor regression. In addition to the genes from the B-F/B-L locus, B-G genes seem to play a major role in tumor regression [86,87].

Avian leukosis virus (ALV) tumors have been reported to be suppressed in birds of the B2 haplotype, whether in a homozygous or a heterozygous state. In contrast, B5 congenic chicken lines have been associated with susceptibility to avian leukosis and a general deficiency in antiviral response [88]. Haplotypes B8a, B9a, and B21 have similar susceptibility to congenital avian leukosis, but differ in susceptibility to horizontal infections [89].

### 3.4. Other Viruses

In field settings, survival to highly pathogenic avian influenza (HPAI) has been associated with the MHC B haplotype B21, whereas high mortality has been associated with haplotype B13 [90,91]. When testing avian influenza disease outcome in several MHC B haplotypes (B2, B12, B13, B19, and B21), differences were observed among different haplotypes, but all groups showed mortality rates between 40% and 100%. The disease outcome varied in B2 haplotype birds of different genetic background, suggesting that non-MHC genes might have a bigger influence on disease severity than the MHC locus [90]. In experiments using avian influenza virus (AIV)-infected macrophages from B2 and B19 haplotype chickens primed in vivo with a plasmid AIV vaccine, it was demonstrated that the B2 haplotype is more efficient at promoting T-cell activation than B19. The resistance feature of B2 seems to be largely influenced by macrophages, which consequently promote a more effective innate immune response that appropriately activates the adaptive immunity [9].

Variation in CD4^+^ T cell responses were observed when different MHC haplotype chickens were challenged with a lentogenic strain of Newcastle disease virus (NDV). Antigen-specific responses of T cells from B12 haplotype chickens were overall low, whereas high responses were observed only in a few individuals of each haplotype [92]. B13 haplotype chickens were associated with greater antibody production than B21 after immunization with NDV [93].

MHC-linked genetic resistance to infectious laryngotracheitis virus (ILTV) has been studied since the 1990s. Homozygous B2 haplotype birds were found to show higher resistance to ILTV infections than heterozygous B15/B21 and B2/B15 haplotype chickens. B2 birds were capable of mounting an immune response with a smaller infectious dose than other tested chicken lines [94]. In a recent study, MHC congenic chicken lines bearing homozygous B2, B5, B12, B13, B19, and B21 haplotypes were challenged with ILTV to evaluate disease incidence. B2 and B5 haplotypes demonstrated greater resistance than the others by showing less severe clinical signs and reduced viral load. Two different lines (lines 6 and 7) containing the B2 haplotype, but a different genetic background, were compared. Line 6 demonstrated greater resistance than line 7, implying the role of non-MHC genes in genetic resistance to this virus [95].

When studying MHC-linked genetic resistance to infectious bursal disease virus (IBDV) using B haplotype homozygote chickens, it was noted that the antibody response to IBDV is MHC II-restricted and T-cell-dependent [96,97]. In a study to test the protection efficacy of a vectorized fowl pox–IBDV vaccine, chicken lines bearing the B haplotypes B2, B12, and B15 were vaccinated and challenged with a virulent IBDV strain, and the bursas were assessed. It was found that B15 does not confer any protection against IBDV, whereas B2 and B12 conferred protection to vaccinated and infected birds [98].

Genetic resistance to viruses and other infectious agents have been thoroughly studied and reviewed [19,62,99,100]. In this brief review, the main goal was to highlight the advances and discoveries that have contributed to the research on MHC-linked genetic resistance to IBV.

## 4. Genetic Resistance and Immune Responses to Infectious Bronchitis Virus Infections

The first evidence of genetic resistance to IBV was reported in 1966, when Purchase et al. used embryos of three chicken lines, namely, line 6 (B2), line 7 (B2), and line 15I (B15), to compare embryonic mortalities after in ovo inoculation with a respiratory strain of IBV. Despite having the same homozygote B2 haplotype, the embryo mortality of line 7 occurred significantly later than in line 6, which was an early indication that background non-MHC genes could be partially responsible for disease severity. Line 15I was considered of intermediate resistance to IBV [101].

A few decades later, challenge experiments using IBV in combination with *E. coli* were performed to study resistance to IBV. The IBV—*E. coli* challenge model facilitated the initial studies on IBV resistance and its inheritance by aggravating the severity of infectious bronchitis with an associated bacterial infection [38,102]. However, it also confounded the ability to detect specific effects of the MHC on IBV infections. Later on, challenge experiments using either a pool of IBV strains or the Massachusetts M41 (M41) IBV strain alone were performed [39,103]. Although both lines were vulnerable to infectious bronchitis, haplotype B12 (line C) was considered resistant, while haplotype B15 (line 15I) was considered susceptible when assessing the duration of the disease, the body weight gain recovery time, and the virus titers over time [103]. In in vivo challenge experiments with IBV, the respiratory tract epithelium of line C recovered more rapidly than that of line 15I. However, in an in vitro IBV challenge, primary cultures of the trachea, lung, liver, and kidney from lines C and 15I seemed equally susceptible, suggesting that the genetic resistance difference between the lines was associated with immune responses and not due to inherent differences in tissue susceptibilities per se [39]. Using tracheal organ cultures (TOCs) derived from C and 15I lines, it was demonstrated by immunohistochemistry (IHC) that IBV M41 replicates for longer periods in line 15I than line C; however, 15I unexpectedly presented with a larger number of immunoglobulin-bearing cells in the trachea compared to C line chickens. This greater number of plasma cells in the susceptible 15I chicken line tracheas may have been a consequence of a severe and damaging initial inflammatory response [104]. Another study showed that bursectomized line C chickens were more resistant to mortality due to IBV infections than intact susceptible 15I chickens, demonstrating that antibodies are not the only component involved in recovery from IBV infections [105]. When further investigating the role of antibodies in lines C and 15I, it was noted that there was no difference in antibody concentration in tracheal washes or serum from these two lines. However, IBV-specific immunoglobulin G (IgG) was detected in greater amount in the saliva and tears of birds from line C (resistant) than those of line 15I (susceptible), suggesting that local humoral responses might be relevant in disease resistance [106].

After a pause of almost a decade, genetic resistance to IBV again became a topic of interest in the 2000s. The nephropathogenic N1/62 IBV strain was used in challenge experiments using two inbred lines of white Leghorn chickens, S and W. In the S line, relatively high mortality was observed with a low 2 × 10^2^ ciliostatic dose 50% (CD_50_). Despite presenting clinical signs of nephritis when challenged with a high dose of nephropathogenic IBV (2 × 10^4^ CD_50_), W line chickens did not exhibit significant mortality and were considered relatively resistant to this IBV challenge [41].

Bacon et al. (2004) [42] reported a field event in which four flocks of breeder chicks were mistakenly vaccinated with a moderately attenuated Mass-type vaccine intended for older chickens. The vaccinated flocks involved two inbred lines, 7_1_ (B2) and 15I_5_ (B15), and two lines congenic to 15I_5_, bearing haplotypes B13 (15.P-13) and B21 (15.N-21). The inbred line 15I_5_ showed greater resistance than its congenic counterparts B13 and B21 lines and inbred line 7_1_ [42], which contradicts previous findings of line 15I being susceptible to IBV [39,103,104,105,106]. Furthermore, a study evaluating commercial heterozygous chickens with haplotypes B2/B15 and B2/B21 vaccinated and challenged with Ark-type IBV strains demonstrated that the line bearing the B15 haplotype had a lower incidence of respiratory signs than the B2/B21 line. However, other parameters such as histopathology, severity, and duration of disease were not significantly different between the tested MHC genotypes [43], suggesting a possible B2 dominance over B15 and B21 haplotypes. When evaluating B2, B5, B8, B12, and B19 haplotypes in challenge experiments using IBV strains Gray and M41, Banat et al. (2013) reported that B2, B5, and B8 haplotypes were relatively resistant while B12 and B19 were relatively susceptible. In addition to the homozygous chicken lines, a heterozygous group of birds containing the resistant B2 and the susceptible B12 haplotypes showed resistance to IBV, suggesting a dominant effect of B2 over B12 [44], supporting the findings from Joiner et al. (2007) [43].

Similarly to the work from Banat and collaborators [44], our group at the University of California, Davis, studied a wide spectrum of MHC B haplotypes using congenic lines developed by Abplanalp in the 1980s [22,46]. Haplotypes BO, B2, B15, B17, B18, B19, B21, and B24 were assessed in a set of experiments using IBV M41 as the infectious challenge. While B2 and B18 were considered relatively resistant haplotypes, B19 was the most susceptible. Two B21 lines, UCD 336 and UCD 330, were tested; while the former was relatively susceptible to IBV, the latter was moderately resistant [46], suggesting a role of non-MHC genes in disease resistance as previously hypothesized [101]. In our experiments, there was no significant correlation among the number of macrophages, CD4^+^ and CD8^+^ T cells, and resistance to IBV infection [46]. Using polyI:C to mimic an RNA virus infection of macrophages in vitro, Dawes et al. (2014) observed that genetic resistance to viral infections might be directly linked to an intense innate immune response, possibly related to macrophage function rather than macrophage numbers [8]. Conversely, flow cytometry from peripheral blood derived from congenic chickens bearing haplotypes B2, B12, B14, B15, B19, and B21 showed a strong effect of the B haplotype on the number of monocytes/macrophages, as well as CD4^+^ and CD8^+^ T cells. The quantification of these leucocyte subsets correlated with IBV viral load, which overall was low in B2, B15, and B21 haplotypes and high in B19, B12, and B14 [107].

Using UCD 331 (B2) and UCD 335 (B19) as an IBV resistant-susceptible pair of animal models, studies were performed to better understand the mechanism underlying genetic resistance to IBV [12,47]. In an attempt to elucidate the role of innate immune responses in resistant and susceptible birds challenged with two IBV genotypes, RNA sequencing was performed to assess gene expression in tracheas of B2 and B19 chickens [47]. The susceptible B19 line showed a high expression of genes related to inflammation and innate immunity, which potentially explains the more severe clinical signs and tracheal lesions in B19 compared with B2 [12,47]. Meanwhile, B2 birds highly expressed genes related to common cell responses such as cell adhesion, motility, proliferation, differentiation, and survival. Thus, while B2 is able to effectively control the IBV infection, B19 seems to have an exacerbated and damaging inflammatory response [47]. These results were similar to what was observed by Nakamura et al. (1991) when studying differences between IBV-resistant (C) and -susceptible (15I) lines, in which an exaggerated inflammatory response might have induced more severe tracheal damage in the susceptible line [104]. Although no differences in antibody levels were observed in sera from the two chicken lines, B2 chickens presented with higher IgG and IgA levels in tears than B19 chickens after challenges with IBV M41 and ArkDPI [46,47]. The detrimental inflammation elicited in B19 birds in early infection might have interfered with the development of an appropriate local adaptive immune response [47]. An experiment with TOCs revealed that B19 birds seem to present increased baseline proinflammatory cytokine levels even before the IBV challenge, which intensifies the inflammation after IBV infection and, consequently, their susceptibility to more severe disease outcomes compared with the resistant B2 haplotype chickens [12].

Over the past five decades or more, genetic resistance to IBV has been a major research topic that continues to be relevant, yet not fully understood. Table 1 highlights the major haplotype associations with infectious bronchitis resistance and susceptibility that have been made over the years. In some instances, different research groups found similar or equivalent results that support concurring hypotheses. In other instances, research findings were divergent but still able to demonstrate the partial effect of the B haplotype at different levels on disease outcome. Nevertheless, these results show that non-MHC genes linked to innate immunity and inflammation play a relevant role in disease resistance.

## 5. The Future of Research on Genetic Resistance to IBV

Even though research has been done on the role of monocytes and macrophages in genetic resistance [8,9,107], this topic is far from being completely understood and raises several questions that can potentially be elucidated in further research. In vitro and ex vivo functional studies have been used for decades. Currently, a wide range of cell markers and antibodies are available to isolate and study specific cell subsets that might bring to light the differences between resistant and susceptible birds at a cellular and molecular level. In addition, the development of molecular techniques at a low cost to assess gene expression and the unified efforts in sequencing the chicken chromosome 16, aligned with the ease of performing MHC typing, are promising factors contributing to innovative research on genetic resistance.

Our results have suggested that, in addition to the impact of the MHC on genetic resistance, other genes might confer resistance to IBV challenges. Large numbers of innate immune genes such as Toll-like receptor 3 (TLR3), complement component 1 (C1R and C1S), interferon regulatory factor 7 (IRF-7) [45], C-type lectin like receptor 1 (CLEC1), and chemokines (CCL17, XCL1 and CCL1) [47] have been found to be upregulated in susceptible chicken lines [45,47]. These results can be explained by the polygenic nature of disease resistance and the effects of transcriptional regulation that is detached from the genomic determinants per se [47]. In contrast, resistant birds highly express MHC B genes [45], but do not seem to upregulate a large number of genes related to proinflammatory responses [45,47]. Further research using differential gene expression comparing uninfected and IBV-challenged birds of various MHC haplotypes are necessary to validate these findings and investigate in depth the pathways that might be involved in genetic susceptibility differences to IBV.

The chicken Mx proteins induced by an infection with avian influenza virus are able to block an early step of the viral replication cycle [108]. The chicken Mx gene is highly polymorphic, and its antiviral activity is dependent on having an asparagine amino acid at the residue 613 [109]. Inconsistent results on the Mx gene association with genetic resistance have been reported [110,111], which may be explained partly by differences in virus strains used in these experiments, particularly in the virus nucleoprotein gene, the viral protein responsible for determining Mx sensitivity [112]. Nevertheless, exploring the relationship of the chicken Mx gene with genetic resistance to IBV is a promising topic.

In addition to experimenting with homozygous MHC haplotypes using inbred and congenic chicken lines, a better understanding of MHC genotypes present in commercial flocks of layer and broiler chickens might be valuable to fully exploit the bird’s immunological potential. Some particular MHC haplotypes seem to better respond to vaccines [42,43], and this might be a useful tool to enhance prevention of infectious bronchitis in field settings.

## 6. Conclusions

The chicken MHC B locus has been widely studied and continues to be of major interest to immunologists and poultry scientists. Several MHC B haplotypes have been linked with genetic resistance or susceptibility to infectious diseases. The genetic resistance partly conferred by the chicken MHC B locus is an additional tool in the prevention of infectious diseases. This inherent feature is particularly desirable in infectious bronchitis infections because serum antibodies elicited by IBV do not always confer protection. In addition to the MHC and its linked genes, other immune-related genes in chromosome 16 may be involved in the mechanisms underlying such genetic resistance. Innate immunity and inflammation are thought to play a major role in different susceptibilities to IBV, either by promoting an effective cellular and humoral response in resistant birds or by inducing damaging inflammatory responses that hinder an appropriate adaptive immune response in susceptible chickens. Nevertheless, functional studies and molecular investigations are needed to further elucidate genetic resistance to infectious bronchitis and other infectious diseases of chickens.

## Figures and Tables

**Figure 1 vaccines-08-00637-f001:**
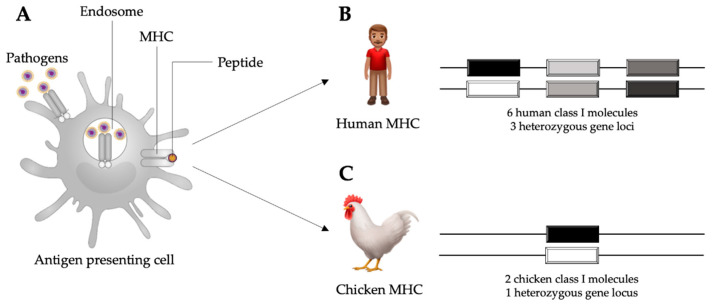
Schematic representation of an antigen presenting cell internalizing pathogens in the process of phagocytosis and subsequent antigen presentation through major histocompatibility complex (MHC) molecules (**A**). In humans, there are six MHC class I molecules occurring in three heterozygous loci (**B**). In chickens, there are two MHC class I molecules occurring in one heterozygous locus (**C**), which allows for stronger associations with disease resistance than the human MHC class I. Adapted from Kaufman, 2013 [19].

**Figure 2 vaccines-08-00637-f002:**
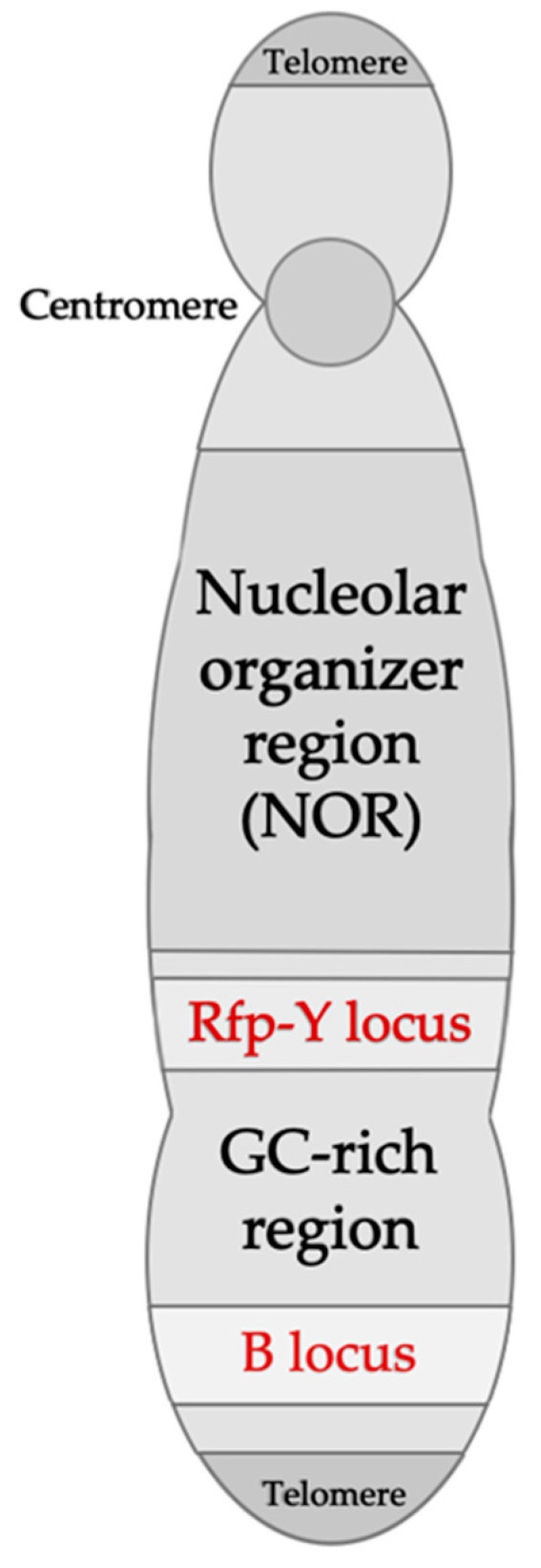
Schematic representation of the chicken microchromosome 16, indicating the MHC B and Rfp-Y loci. The B locus contains the B-F/B-L and the B-G regions corresponding to MHC class I, II, III, and IV (B-G) genes. The Rfp-Y locus contains nonclassical MHC class I and II genes. Adapted from Delany et al. (2009) [61] and Kaufman (2013) [19].

**Table 1 vaccines-08-00637-t001:** Major histocompatibility complex B haplotypes and their association with genetic resistance to infectious bronchitis virus.

Haplotype	Chicken Line	Characteristics	Reference
B2	Line 6	Susceptible	Purchase et al. 1966 [101]
	Line 7	Resistant	Purchase et al. 1966 [101]
		Susceptible	Bacon et al. 2004 [42]
	NIU ^1^	Resistant	Banat et al. 2013 [44], Dawes et al. 2014 [8]
	UCD ^2^ 331	Resistant	da Silva et al. 2017 [46], da Silva et al. 2019 [47], da Silva et al. 2020 [12]
	AU ^3^	Resistant	Larsen et al. 2019 [107]
B5	NIU	Resistant	Banat et al. 2013 [44]
B8	NIU	Resistant	Banat et al. 2013 [44]
B12	Line C	Resistant	Cook et al. 1990 [39], Otsuki et al. 1990 [103], Nakamura et al. 1991 [104], Cook et al. 1991 [105] Cook et al. 1992 [106], Smith et al. 2015 [45]
	NIU	Susceptible	Banat et al. 2013 [44]
	AU	Susceptible	Larsen et al. 2019 [107]
B13	15.P-13	Intermediate	Bacon et al. 2004 [42]
B14	AU	Susceptible	Larsen et al. 2019 [107]
B15	AU	Resistant	Larsen et al. 2019 [107]
	15I	Resistant	Bacon et al. 2004 [42]
		Intermediate	Purchase et al. 1966 [101]
		Susceptible	Cook et al. 1990 [39], Otsuki et al. 1990 [103], Nakamura et al. 1991 [104], Cook et al. 1991 [105] Cook et al. 1992 [106], Smith et al. 2015 [45]
B18	UCD 253	Resistant	da Silva et al. 2017 [46]
B19	NIU	Susceptible	Banat et al. 2013 [44], Dawes et al. 2014 [8]
	UCD 077	Susceptible	da Silva et al. 2017 [46]
	UCD 335	Susceptible	da Silva et al. 2017 [46], da Silva et al. 2019 [47], da Silva et al. 2020 [12]
	AU	Susceptible	Larsen et al. 2019 [107]
B21	AU	Resistant	Larsen et al. 2019 [107]
	15.N-21	Intermediate	Bacon et al. 2004 [42]
	UCD 330	Susceptible	da Silva et al. 2017 [46]
	UCD 336	Susceptible	da Silva et al. 2017 [46]

^1^ NIU = Northern Illinois University; ^2^ UCD = University of California, Davis; ^3^ AU = Aarhus University.

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
