# Peer review of "The Chicken MHC: Insights into Genetic Resistance, Immunity, and Inflammation Following Infectious Bronchitis Virus Infections"

_vaccines, 2020, doi:10.3390/vaccines8040637_

Round 1

Reviewer 1 Report

  In this review, the authors summarized the significance of the chicken major histocompatibility complex (MHC) and its impact on poultry disease resistance, particularly in infectious bronchitis virus (IBV) infections, and the future of IBV genetic resistance research using MHC inbred and congenic chicken lines.
  The review is interesting and meaningful, but it needs some minor revisions.

  1. Check that all acronyms used in the manuscript are carefully defined.
  2. Add a comparison between immunity in humans and that of chicken. What is the difference between these two immune responses?
  3. There are some typos. The authors should carefully read the manuscript.
  4. Check rigourously the references used in this review.

Author Response

REVIEWER 1

 In this review, the authors summarized the significance of the chicken major histocompatibility complex (MHC) and its impact on poultry disease resistance, particularly in infectious bronchitis virus (IBV) infections, and the future of IBV genetic resistance research using MHC inbred and congenic chicken lines.

  The review is interesting and meaningful, but it needs some minor revisions.

  1. Check that all acronyms used in the manuscript are carefully defined.

The authors checked all the acronyms in the manuscript for proper definition.

  1. Add a comparison between immunity in humans and that of chicken. What is the difference between these two immune responses?

The major differences between the avian and the mammalian immune systems were presented and expanded in lines 36-43.

  1. There are some typos. The authors should carefully read the manuscript.

The manuscript was reread and typos were fixed throughout the manuscript.

  1. Check rigorously the references used in this review.

The references were checked, and more references were added.

Reviewer 2 Report

Interesting and well structured review on avian MHC

Author Response

Interesting and well-structured review on avian MHC.

The authors appreciate the reviewer’s feedback.

Reviewer 3 Report

The authors of "The chicken MHC: Insights on genetic resistance, immunity and inflammation following infectious bronchitis virus infections " present a review on the current information available on genetic resistance (focusing on MHC in chickens ) to infectious bronchitis virus infections.

The authors have written a comprehensive helpful review on the state of our understanding of genetic resistance in chickens to infectious agents.

General Comments:

The review needs a senior English writer to fully edit the manuscript focusing on making the writing concise and clear. The writing, especially at the beginning of the review, needs to be more specific and less general.

Given the audience of this journal, it would be appropriate to add a segment or expand our understanding of how T cell or cell-mediated immunity plays a role in protecting chickens from infectious diseases, focusing on infectious bronchitis virus.

Specific Comment:

The writing will be clearer if it was more direct.

A few examples,

Line 30 The sentence is awkward and almost suggests that studying the avian immune system has provided insight into research itself. Please clarify

line 34, The statement highlights the bursa of fabricius function but never clearly states the “function”. To most it may seem obvious but there is a pattern of not directly saying what you mean, which can be written in half a sentence or just eliminating the word function and stating it right away.

The authors state numerous times that the chicken MHC is small and simple. This is very unclear. For example,

Line 41 what does “the chicken MHC is 20-fold smaller than the mammalian MHC” mean?

Figure 1 is helpful, but incomplete. The authors state the MHC is less complex and mention that the mammalian MHC is more complex, but this is a review and should have hard numbers to highlight the differences. For instance, how many alleles are present in the different human MHC loci? How many alleles exist for the chicken equivalent? Does less complex or small have to do mostly with having a single locus in chicken? Are there fewer alleles at the locus?

Clarifying what the MHC means is always useful. Line 140 the authors write “MHC genotype”, and later the authors write “MHC haplotypes”. Line 123, do the authors mean MHC haplotypes?

Line 164 The authors bring up the idea of the interplay between vaccine serotype and MHC haplotype citing four studies and then ignore it. The authors must expand and explain this interplay, which would be of interest to the audience.

Line 168 do the authors mean “located” instead of "allocated"?

Line 285 unclear what “counts” the authors are referring to. The authors do not write any numbers nor do they explain how they are able to make “counts” using flow cytometry.

Line 295 unclear what “cell responses” refers to

Line 297 restate the observation by Nakamura

Line 319 do you mean raise instead of “arise”

Line 333 Unclear what the sentence beginning with “On the other hand,” is saying

Author Response

REVIEWER 3

The authors of "The chicken MHC: Insights on genetic resistance, immunity and inflammation following infectious bronchitis virus infections " present a review on the current information available on genetic resistance (focusing on MHC in chickens) to infectious bronchitis virus infections.

The authors have written a comprehensive helpful review on the state of our understanding of genetic resistance in chickens to infectious agents.

General Comments:

The review needs a senior English writer to fully edit the manuscript focusing on making the writing concise and clear. The writing, especially at the beginning of the review, needs to be more specific and less general.

The manuscript was reviewed by a native English speaker for clarity.

The introductory portion of this review was written with a broader perspective to facilitate the read. The focus of this review is the research on genetic resistance to IBV. In the IBV section, the writing was very specific. The introduction was expanded as shown in lines 31 to 63.

Given the audience of this journal, it would be appropriate to add a segment or expand our understanding of how T cell or cell-mediated immunity plays a role in protecting chickens from infectious diseases, focusing on infectious bronchitis virus.

A paragraph on cell-mediated immunity was added in lines 52 to 63.

Specific Comment:

The writing will be clearer if it was more direct.

A few examples,

Line 30 The sentence is awkward and almost suggests that studying the avian immune system has provided insight into research itself. Please clarify

The purpose of this phrase was to highlight the importance of the avian animal model in generating knowledge that serve as a basis for major general immunology concepts we use nowadays. The referred sentence was rephrased in lines 31-32.

line 34, The statement highlights the bursa of fabricius function but never clearly states the “function”. To most it may seem obvious but there is a pattern of not directly saying what you mean, which can be written in half a sentence or just eliminating the word function and stating it right away.

The suggested modification was done in lines 34-35.

The authors state numerous times that the chicken MHC is small and simple. This is very unclear. For example,

Line 41 what does “the chicken MHC is 20-fold smaller than the mammalian MHC” mean?

The size difference between the human and the chicken MHC was first described by Kaufman et al in 1999 (https://www.nature.com/articles/44856) and later on several other publications by Kaufman and collaborators. They describe the genomic size difference exactly as follows: “the B locus contains only 19 genes, making the chicken MHC roughly 20-fold smaller than the human MHC”. To clarify and provide more up-to-date information, we changed the referred sentence in lines 66-68.

Figure 1 is helpful, but incomplete. The authors state the MHC is less complex and mention that the mammalian MHC is more complex, but this is a review and should have hard numbers to highlight the differences. For instance, how many alleles are present in the different human MHC loci? How many alleles exist for the chicken equivalent? Does less complex or small have to do mostly with having a single locus in chicken? Are there fewer alleles at the locus?

The number of genes and base pairs of the human and chicken MHC are in lines 64 to 66. Class I molecules on the human MHC contain two alleles of each HLA-A, HLA-B and HLA-C, whereas the chicken MHC contains two alleles of the BF2 locus as exemplified in Figure 1. The aim of this review is to summarize the effects of resistance to infectious disease, particularly to infectious bronchitis, and provide a simple and easy-to-follow background on the genetic aspects underlying such resistance to a non-geneticist audience.

Clarifying what the MHC means is always useful. Line 140 the authors write “MHC genotype”, and later the authors write “MHC haplotypes”. Line 123, do the authors mean MHC haplotypes?

Yes. The word “haplotype” was added in lines 149 (former 123) and 165 (former 140).

Line 164 The authors bring up the idea of the interplay between vaccine serotype and MHC haplotype citing four studies and then ignore it. The authors must expand and explain this interplay, which would be of interest to the audience.

We respectfully disagree from the reviewer’s comment. The authors did not ignore the citations. All five references are from the same research group and conclude that for the induction of maximum Marek’s disease resistance, selecting an MDV vaccine strain that is appropriate for the predominant B haplotypes of the flock is ideal. This conclusion was summarized in our review as “tailored vaccine strains should be used when certain MHC haplotypes are predominant in a population”. An example of such observation was added in lines 188 to 192.

Line 168 do the authors mean “located” instead of "allocated"?

The word “allocated” was substituted by the word “located” (line 195).

Line 285 unclear what “counts” the authors are referring to. The authors do not write any numbers nor do they explain how they are able to make “counts” using flow cytometry.

The word “counts” was substituted by the word “numbers” (line 313). Flow cytometric cell counting protocols are out of the scope of this manuscript.

Line 295 unclear what “cell responses” refers to

“Cell responses” refer to ordinary cellular responses not necessarily related with immunity or inflammation, such as cell adhesion, motility, proliferation, differentiation and survival. This information was added in line 323-324.

Line 297 restate the observation by Nakamura

The observation by Nakamura et al (1991) was added in lines 327-328.

Line 319 do you mean raise instead of “arise”

The word “arise” was substituted by the word “raise” (line 351).

Line 333 Unclear what the sentence beginning with “On the other hand,” is saying

The sentence was rephrased (lines 365-367).

Round 2

Reviewer 3 Report

The authors of "The chicken MHC: Insights on genetic resistance, 2 immunity and inflammation following infectious 3 bronchitis virus infections" have made significant changes that clarify their review.